# Optical Design and Optimization with Genetic Algorithm for High-Resolution Optics Applied to Underwater Remote-Sensing

Chun-Feng Chou [1], Cheng-Mu Tsai [2], Chao-Hsien Chen [1], Yung-Hao Wong [3], Yi-Chin Fang [1,*], Chan-Chuan Wen [1], Hsiao-Yi Lee [1], Hien-Thanh Le [4], Shun-Hsyung Chang [1] and Hsing-Yuan Liao [1]

1    Department of Mechanical Engineering, National Kaohsiung University of Science and Technology, Kaohsiung City 811, Taiwan; F107142116@nkust.edu.tw (C.-F.C.); ch_chen@nkust.edu.tw (C.-H.C.); wencc@nkust.edu.tw (C.-C.W.); leehy@nkust.edu.tw (H.-Y.L.); shchang@nkust.edu.tw (S.-H.C.); Sindy@nkust.edu.tw (H.-Y.L.)
2    Graduate Institute of Precision Engineering, National Chung Hsing University, Taichung City 402, Taiwan; jmutsai@email.nchu.edu.tw
3    Department of Mechanical Engineering, Minghsin University of Science and Technology, Hsinchu 30401, Taiwan; yvonwong@must.edu.tw
4    Department of Technology, Dong Nai Technology University, Bien Hoa 810000, Vietnam; lethanhhien2012@gmail.com
*    Correspondence: yfang@nkust.edu.tw

**Abstract:** In fields such as biology, archeology, and industry, underwater photogrammetry can be achieved using consumer-grade equipment. However, camera operations underwater differ considerably from those on land because underwater photogrammetry involves different optical phenomena. On the basis of the requirements and specifications of the marine vessel Polaris, we developed a novel underwater camera with prime and zoom lenses and a high resolving power. The camera can be used in the spectrum in shallow water and the blue–green spectrum in deep water. In the past, ordinary cameras would be placed in waterproof airtight boxes for underwater photography. These cameras were not optimized to the underwater spectrum and environment, resulting in no breakthroughs in resolving power. Furthermore, the use of the blue spectrum greatly increases during underwater and particularly deep-water surveying. Chromatic aberration and focus-point displacement generated by the shift from the shallow-water spectrum to the blue–green spectrum in deep water makes universal underwater photography even more difficult. Our proposed optical design aimed to overcome such challenges for the development of a high-resolution underwater surveying camera. We designed a prime lens and a zoom lens. We adopted a waterproof dome window on the outer surface as the basic structure and optimized it in accordance with the conditions of different water depths and spectra to obtain distortion within ±2% and high-resolution underwater imaging quality. For the zoom lens design, we employed a genetic algorithm in Zemax to attenuate chromatic aberration as a kind of extended optimization. This novel optical design that can be used in all waters is expected to greatly reduce the volume and weight of conventional underwater cameras by more than 50% and 60%, respectively, and increase their resolving power by 30–40%.

**Keywords:** optical design; photogrammetry; extended optimization; light spectrum; genetic algorithm

## 1. Introduction

Underwater photogrammetry, similar to terrestrial and aerial photogrammetry, has advanced with improvements in photography. Applications of underwater photogrammetry are increasing because of technical improvements to photographic equipment, underwater manned and unmanned vehicles, and diving apparatus. Many scientists have used underwater cameras to explore the underwater world and observe marine animals. McNeil elaborated on the structure of underwater cameras (see Figure 1) and identified parameters

for ocean optics [1]. An underwater lens is composed of a dome window and an air lens assembly. Nikon Corporation obtained a patent for a wide-angle underwater lens in 1996 [2]. The robotics group of Łuczyński and Birk [3] introduced various situations in which haze interferes with underwater images.

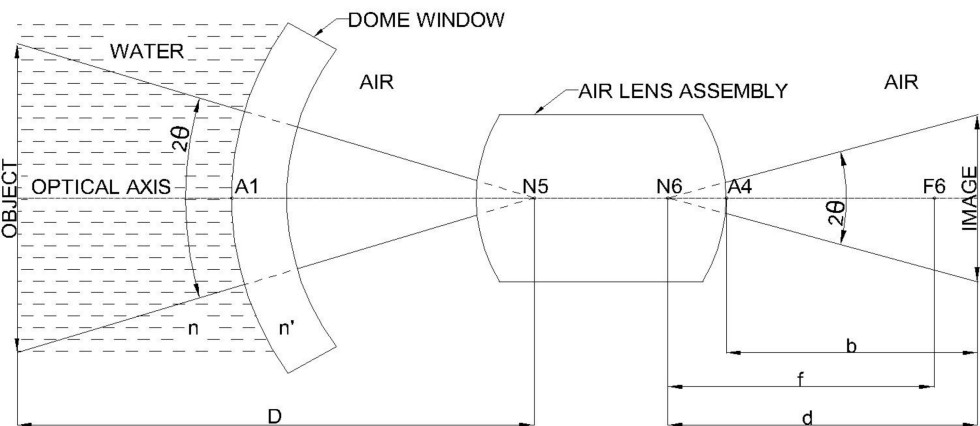

**Figure 1.** Underwater camera [1]: Photo from 1. Gomer T. McNeil, *Metrical Fundamentals of Underwater Lens System*, Optical Engineering, vol. 16, no. 2, pp. 128–139 (1977).

The refractive index of water changes with salinity, temperature, pressure, and optical wavelength. The intensity of different colors of light varies at different water depths because of light absorption and attenuation. The penetration of different colors of light also varies because of different optical wavelengths. Accordingly, optical aberration greatly reduces the resolving power of conventional optical cameras in waterproof boxes during marine surveying [4].

To address these problems, we designed an optical camera based on full and blue-green spectra and optimized it according to underwater conditions; this optimization is essential for enhancing the resolving power of underwater cameras. From the early exploration of underwater imaging technology [5,6] to the latest research, few researchers have made breakthroughs regarding the fundamental problems of optical lenses [7,8]; specifically, extensive progress has been made in waterproofing technology for lenses from the perspective of optical design and aberration [9]. From the perspective of modern optical engineering, technologies with IP67 (industrial optics) or higher waterproof ratings and the technique of inserting inert gas into barrels are relatively mature. The novel optical design for underwater cameras in the present study could be a major breakthrough.

On the basis of the requirements of the research vessel Polaris, we developed an optical design for prime and zoom lenses applicable in the underwater spectrum. First, we removed the conventional outer waterproof box included in the IP67 specification because the glass surface of the box reduces the resolving power of the camera and the penetration of light, and it increases the volume and weight of the camera. Moreover, chromatic aberration and focus-point displacement caused by the shift from the shallow-water spectrum to the blue, deep-water spectrum makes universal deep-water photography even more difficult. These two optical aberrations also reduce the resolving power. In this research, we wrote a program using the genetic algorithm by Matlab language connected with Zemax via its macro language to optimize optics and selected appropriate glass to greatly reduce the influence of chromatic aberration. Using the optimization method via CODEV and Zemax we developed a novel optical zoom lens applicable in all waters to reduce the volume and weight of the camera by more than 50% and 60%, respectively, and increase its resolving power by 30–40% relative to conventional underwater cameras.

## 2. Background

### 2.1. Light Waves

Light can be classified into visible and invisible wavelengths according to human vision, as shown in Figure 2. The wavelengths of visible light range from 380 to 760 nm. Violet, indigo, blue, green, yellow, orange, and red light have wavelengths from short to long. The wavelengths of invisible light are longer than 760 nm and shorter than 380 nm. Visible spectra are adopted in many optical systems, but invisible spectra are also applied in some technologies.

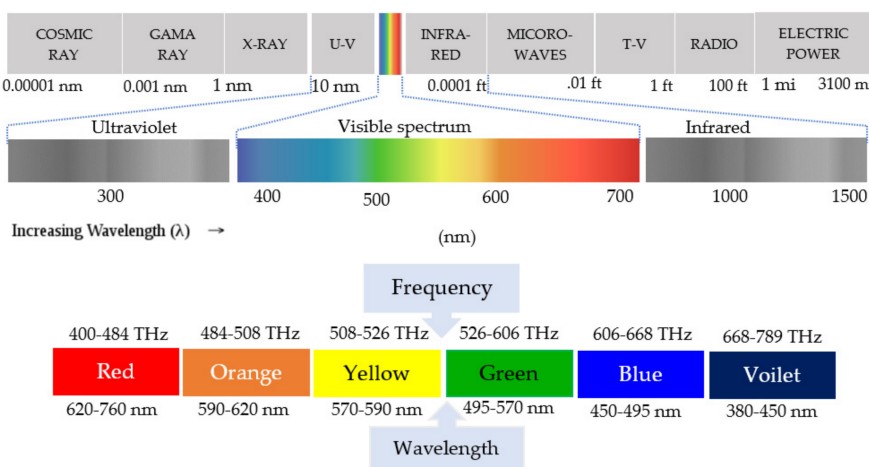

**Figure 2.** General visible spectrum. The wavelength of visible light ranges from 380.00 to 760.00 nm. Violet, indigo, blue, green, yellow, orange, and red light have wavelengths from short to long.

### 2.2. Underwater Light Intensity

Figure 3 was adopted from an article published in an IEEE international conference [10]. The horizontal axis is the depth of water from 0 to 20 m, and the vertical axis is light intensity (%). In general, blue and green lights attenuate the slowest and second slowest in intensity, but red light attenuates the fastest. At 12 m underwater, the intensities of blue, green, and red lights are greater than 60%, approximately 60%, and nearly 0%, respectively. At 20 m underwater, the intensities of blue, green, and red lights are greater than 45%, greater than 40%, and 0%, respectively. Colors of objects observed in underwater environments are different from those in air.

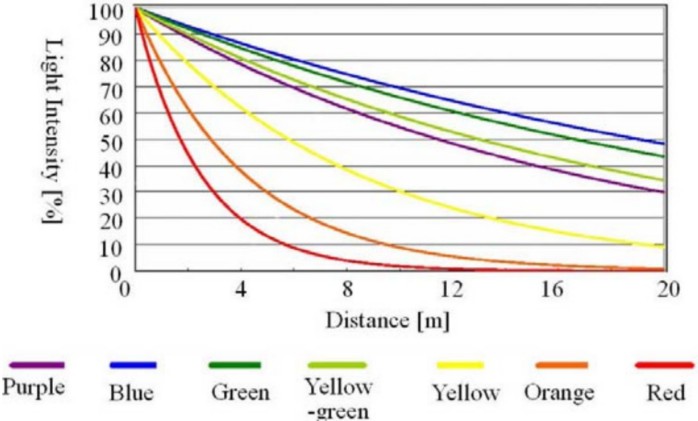

**Figure 3.** Spectrum of the horizontal axis which is around the water depth from 0 to 20 m, and the vertical axis is light intensity (%). Blue and green lights attenuate the slowest and second slowest in intensity; red light attenuates the fastest. This diagram is from reference [10].

Generally speaking, the light intensity decreases with the distance from objects in liquid by light attenuation depending on the wavelength of light. Figure 3 shows the light attenuation in water. For example, the intensity of red color decreases to about a half at the point whose distance from the light source is 2 m, although that of blue color hardly changes. Red color disappears at 20 m distance. In this way, colors of objects in water look different from those in air. The measurement system and its set-up is described in [10].

### 2.3. Penetration of Light Waves in a Lake

Figure 4 depicts the results of sunlight penetration in Lake Superior obtained by the students and scientists at the University of Minnesota Duluth [11]. In this lake, the penetration depth of red light is less than 50 m, whereas that of blue and green lights exceeds 100 m.

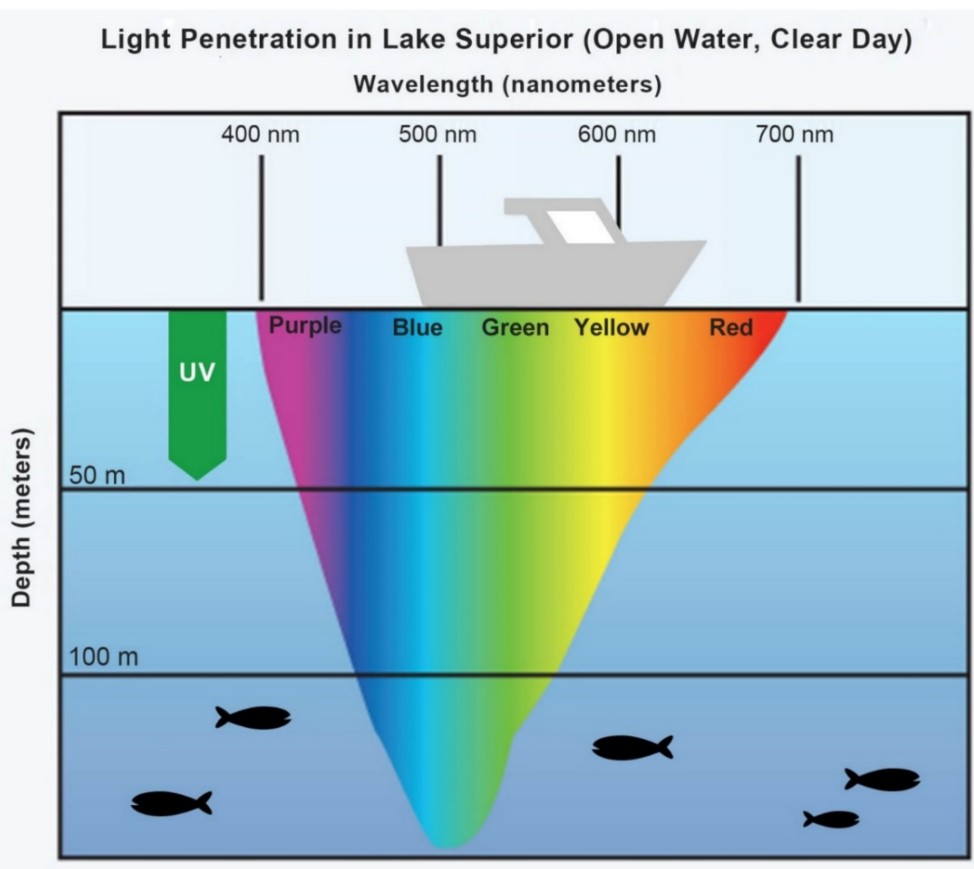

**Figure 4.** Penetration of light at different wavelengths [11]. In Lake Superior, the penetration depth of red light is less than 50 m, whereas that of blue and green lights exceeds 100 m.

### 2.4. Relationship between Wavelength and Refractive Index

As mentioned in Section 2.1, light is divided into visible and invisible spectra. Table 1 presents the refractive indices of light at different wavelengths at a water temperature of 20 °C. A shorter wavelength results in a higher refractive index; conversely, the longer the wavelength, the lower the refractive index.

### 2.5. Variances between Wavelength and Refractive Index

Table 2 presents the changes in the refractive index of yellow sodium light at a wavelength of 589.3 nm in water at different salinity and temperature levels. The unit for salinity is ‰ or ppt (grams of salt per thousand grams of water), and it ranges from 0‰ to 40‰. The unit for temperature is °C and it ranges from 0 °C to 30 °C. The refractive index is approximately 1.3 and varies slightly at different salinity and temperature levels.

**Table 1.** Refractive indices of light at different wavelengths at a water temperature of 20 °C [10].

| Press, Chapter 2 Ocean optics, P.R.C. | | | |
|---|---|---|---|
| **Wavelength (nm)** | **Refractive Index** | **Wavelength (nm)** | **Refractive Index** |
| 250 | 1.3773 | 486 | 1.3371 |
| 308 | 1.3569 | 589 | 1.3330 |
| 359 | 1.3480 | 768 | 1.3289 |
| 400 | 1.3433 | 1000 | 1.3247 |
| 434 | 1.3403 | 1250 | 1.3210 |

**Table 2.** Variables in the refractive index of the D-Line wavelength [10].

| Salinity (‰, ppt) | Temperature (°C) | | | |
|---|---|---|---|---|
| | **0** | **10** | **20** | **30** |
| 0 | 1.33400 | 1.33369 | 1.33298 | 1.33194 |
| 5 | 1.33498 | 1.33463 | 1.33390 | 1.33284 |
| 10 | 1.33597 | 1.33557 | 1.33482 | 1.33374 |
| 15 | 1.33595 | 1.33652 | 1.33573 | 1.33464 |
| 20 | 1.33793 | 1.33746 | 1.33665 | 1.33554 |
| 25 | 1.33892 | 1.33840 | 1.33757 | 1.33644 |
| 30 | 1.33990 | 1.33934 | 1.33849 | 1.33734 |
| 35 | 1.34088 | 1.34028 | 1.33940 | 1.33824 |
| 40 | 1.34186 | 1.34123 | 1.34032 | 1.33914 |

### 2.6. Equation for the Refractive Index of Water

According to McNeil [1], Equation (1) calculates the refractive index in water. The refractive index in water is associated with salinity, temperature, optical wavelength, and pressure, and it can be obtained when these variables are known. In Equation (1), $n_w$ is the refractive index of water; $\lambda$ is the optical wavelength (in nm); $T$ is the water temperature (in °C); $S$ is the salinity (in ‰ or ppt); and $P$ is the pressure (in kg/cm$^2$).

$$n_w = 1.3247 + \frac{3.3 \times 10^3}{\lambda^2} - \frac{3.2 \times 10^7}{\lambda^4} - 2.5 \times 10^{-6}T^2 + \left(5 - 2 \times 10^{-2}T\right)\left(4 \times 10^{-5}S\right) + \left(1.45 \times 10^{-5}P\right)\left(1.021 - 6 \times 10^{-4}S\right)\left(1 - 4.5 \times 10^{-3}T\right) \tag{1}$$

### 2.7. Distortion

In ideal optics, the magnification of each field of view is identical. However, in reality, after light passes through an optical system and is focused on the imaging surface, the magnification differs at different image heights, and the objects in the images are then deformed, as depicted in Figure 5. This is called distortion. In positive distortion, also called pincushion distortion, the actual imaging surface is larger than the ideal imaging surface. In negative distortion, also called barrel distortion, the actual imaging surface is smaller than the ideal imaging surface.

### 2.8. Chromatic Aberration

The refractive index of glass varies depending on its material, whereas that of light differs by its wavelength. After entering an optical system, a ray of white light produces light in different color bands when refraction occurs. Chromatic aberration occurs when different colored light is focused at different positions. Chromatic aberration is positive when the focus point of light with long wavelengths is left of that of light with short wavelengths. Conversely, chromatic aberration is negative when the focus point of the light with long wavelengths is right of that of light with short wavelengths. Generally, red light is longer in wavelength and blue light is shorter in wavelength. In addition, chromatic aberration can be categorized into axial and lateral. As depicted in Figure 6,

axial chromatic aberration indicates that the difference between the focus points of light at different wavelengths is on the optical axis after refraction in an optical system. As illustrated in Figure 7, lateral chromatic aberration refers to a difference between the focus points of light at different wavelengths on the ideal imaging surface after refraction in an optical system.

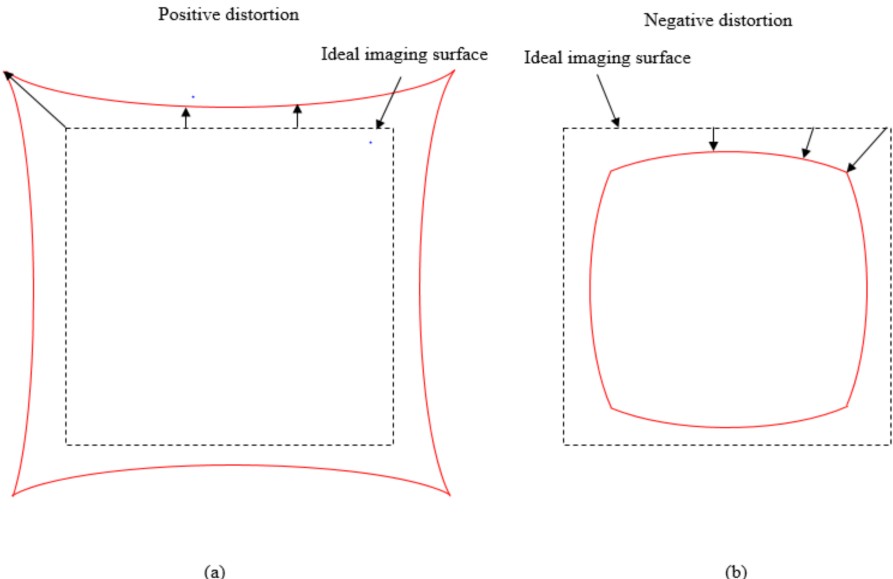

**Figure 5.** (**a**) Positive and (**b**) negative distortion.

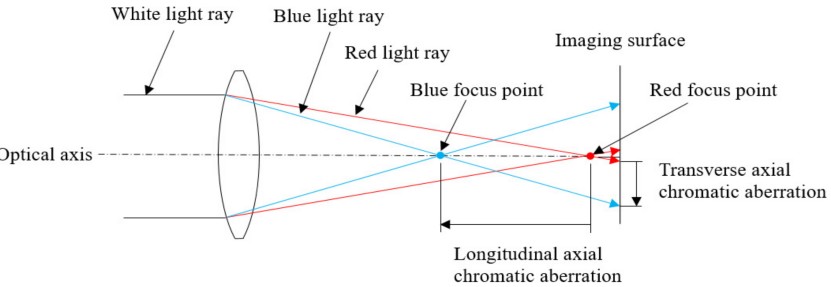

**Figure 6.** Axial chromatic aberration. Such chromatic aberration typically affects the displacement of the focus points of different colored light in different spectra and enables underwater cameras to produce absolute images.

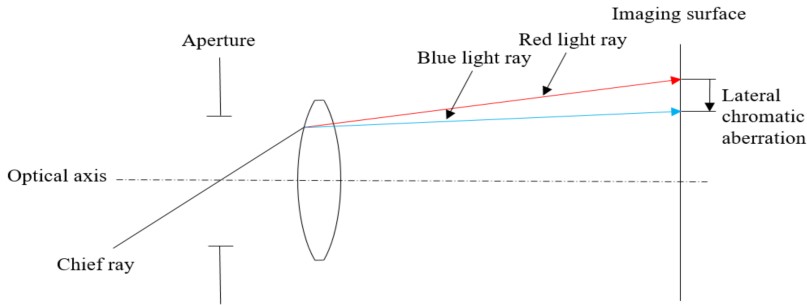

**Figure 7.** Lateral chromatic aberration. Such chromatic aberration typically occurs at the edges of images and is commonly used with a haze to produce an effect similar to flare. Although nonconsequential in this study, the selection of glass using the genetic algorithm increases image quality and reduces flare.

### 3. Brief Introduction to Polaris

Polaris is the product of an industry–academia partnership between Dragon Prince Hydro-Survey Enterprise Co. Kaohsiung City 800, Taiwan and National Kaohsiung University of Science and Technology (Nanzi Campus), and it is Taiwan's first nongovernmental oceanographic research and survey vessel. It was built by Shing Sheng Fa Boat Building Co, Ltd. Kaohsiung City 80544, Taiwan. It weighs 260 tons has a top speed of 11 knots, and is 36.98 m long by 6.80 m wide. Since 2008, Polaris has been stationed at the Innovation Incubation Center of National Kaohsiung University of Science and Technology, and it is typically parked at the jetty of the Cijin Campus. Polaris has completed many underwater exploration and rescue operations. For example, various parties were involved in the search for Malaysia Airlines flight MH370, which mysteriously disappeared on 8 March 2014. The Indian Ocean is approximately 3900 m deep on average, and approximately 8000 m deep at its deepest. The search operations required advanced equipment, and Polaris was invited to Australia to use its underwater detection and deep tow system to help in salvation operations. This demonstrated the international recognition of the high-quality system on Polaris. Photographs and diagrams of the vessel are presented in Figure 8a–c.

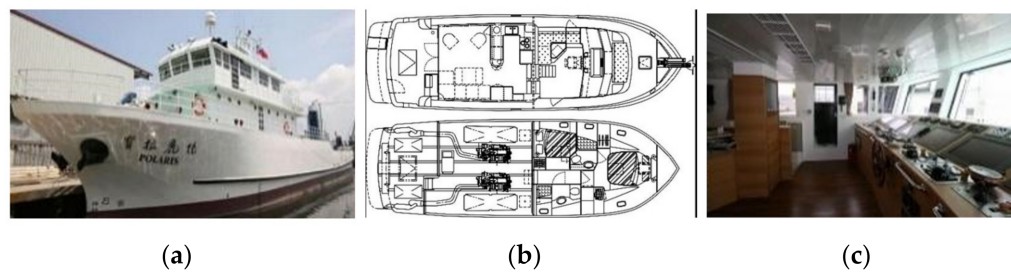

(**a**) (**b**) (**c**)

**Figure 8.** (**a**) Polaris, (**b**) cabin configuration, and (**c**) data-processing center.

Polaris is equipped with a differential satellite positioning system, underwater remotely operated vehicle, single-beam and multibeam echo sounders, subbottom profiler, side-scan sonar system, underwater positioning system, sediment corer, marine magnetometer, and penetrometer.

### 4. Methodology: Optical Design of the Prime System

*4.1. Specifications for Underwater Optical Systems*

The design was based on the specification requirements of Polaris. We first designed the prime lenses followed by zoom lenses. In terms of chromatic aberration, spherical-chromatic aberration, and different spectra, the executive software CODE V was powerful enough to obtain relatively small chromatic aberration when appropriate settings were used. This suggested that we could achieve satisfactory results for prime lens design without using the genetic algorithm.

Underwater lenses are composed of a dome window and an air lens assembly, and the two components should be designed together. The initial architecture of the underwater lenses designed in this study was a dome window and a double-Gauss lens, as illustrated in Figure 9. We used quartz glass as the material for the dome window in consideration of water pressure and Schott lenses for the air lens assembly.

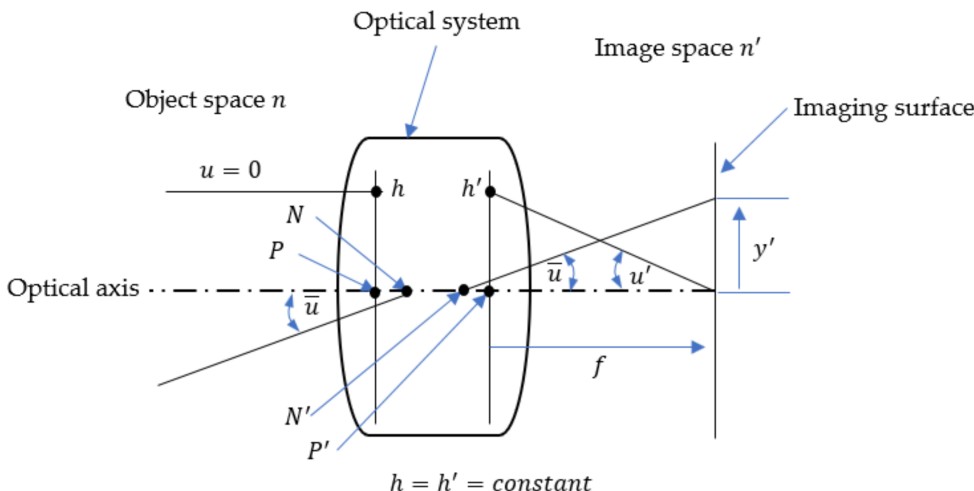

**Figure 9.** Optical system with the object at infinity.

Theoretically, in an optical system where the object is at infinity, the refractive indices of the object space and the image space are different. The refractive index of the object space is $n$, and that of the image space is $n'$. The focal length is $f$, and the image height is $y'$. The first and second principle points are $P$ and $P'$, respectively; the first and second nodal points are $N$ and $N'$, respectively. The incident and emergent angles of the chief ray are $\bar{u}$. The included angle between the marginal ray and the optical axis is $u$ in the object space and $u'$ in the image space. The heights of the incident point and emergent point of the marginal ray are $h$ and $h'$, respectively. By using the optical invariant [12], we could infer the relationship among refractive index, focal length, angle of view, and image height, as expressed in Equation (8) [13].

On the image surface

$$H = n\left(\bar{h}u - h\bar{u}\right) = -nh\bar{u} \tag{2}$$

On the object surface

$$H = n\left(\bar{h}u - h\bar{u}\right) = n'y'u' \tag{3}$$

Equivalent image and object surface according to the optical invariant

$$-nh\bar{u} = n'y'u' \tag{4}$$

Image height

$$y' = \frac{n}{n'}\frac{h}{-u'}\bar{u} \tag{5}$$

Focal length

$$f = \frac{h}{-u'} \tag{6}$$

Image height obtained by substituting Equation (6) into Equation (5)

$$y' = \frac{n}{n'}f\bar{u} \tag{7}$$

Image height in a nonparaxial optical system

$$y' = \frac{n}{n'}f\tan\bar{u} \tag{8}$$

The specifications of the underwater lenses designed for the full and blue–green spectra are listed in Table 3. The actual image height detected by the sensor was 21.6 mm, the pixel size was 7.13 μm, and the spatial frequency was 70 lp/mm. The aperture and focal

length were f/2 and 75 mm, respectively, and a half-angle of view of 12.20° was obtained. The half-angles of view and their weight ratios are listed in Table 4. We selected seven fields of view for the design of the prime lens. The weight of the on-axis field of view was the largest, and that of the outermost field of view was the smallest.

**Table 3.** Specifications of the novel underwater prime lens.

| | |
|---|---|
| Resolution | 4800 × 3600 (1700 K) |
| Sensor size | 36 mm × 24 mm (CMOS) |
| Diagonal length | 43.2 mm |
| Actual image height | 21.6 mm |
| Object distance | Infinity |
| Spatial cutoff frequency | 70 lp/mm |
| Half-angle of view | 12.20° |
| Focal length | 75 mm |
| Aperture | f/2 |

**Table 4.** Fields of view and their weights.

| Half-Angle of View | Weight |
|---|---|
| 0° | 3 |
| 2.2° | 3 |
| 4.5° | 2 |
| 6.0° | 2 |
| 8.4° | 2 |
| 10.8° | 1 |
| 12.2° | 1 |

### 4.2. Design and Verification of the Underwater Optical Camera

The selected wavelengths from the blue–green spectrum are listed in Table 5. We selected green and blue light wavelengths because they have superior penetration ability in lakes. The weight ratio of the three selected wavelengths was 1:1:1, and the center wavelength was in the blue light range. The lens was designed to reduce the chromatic aberration value, particularly that of green and blue light, to a low point. In CODE V, we mainly used wavelength weights to attenuate chromatic aberration.

**Table 5.** Weight ratios for wavelengths in the blue–green spectrum.

| Wavelength | Weight |
|---|---|
| 546.1 nm ($n_e$) | 1 |
| 486.1 nm ($n_F$) | 1 |
| 479.9 nm ($n_{F'}$) | 1 |

As depicted in Figure 10, the lens consists of a dome window and an air lens assembly, and the optical system has 11 lenses in total. In addition to an infinite object distance, various object distances should be considered in photography. Therefore, we conducted tests with object distances of infinity, 10 m, and 5 m.

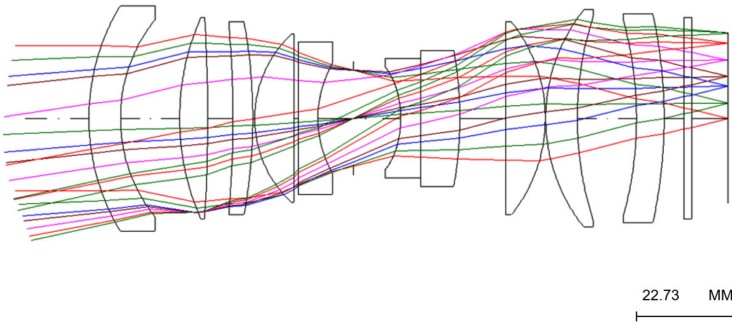

**Figure 10.** Optical design layout of the underwater lenses.

The spot size diagrams in Figures 11–13 show how light rays are focused on the ideal imaging surface. The half-angle of view gradually increases from bottom to top, with the bottommost angle on the optical axis being 0° and the topmost angle being 12.2°. The scales of the diagram are all 0.1 mm. Figures 11–13 present the spot size diagrams with an infinite, 10-m, and 5-m object distance, respectively.

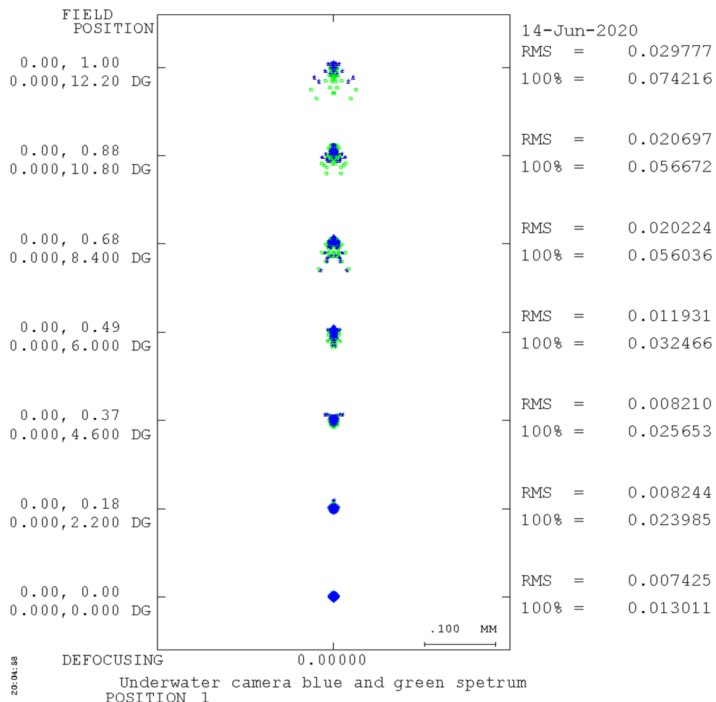

**Figure 11.** Spot size diagram with an infinite object distance (generated by CODE V).

Figures 14–16 present the spherical aberration, astigmatism field curvature, and distortion with an object distance of infinity, 10 m, and 5 m, respectively. The scales of the spherical aberration range from −0.1 to 0.1. The optical axis does not have astigmatism, but the off-axis fields of view do. The distortion of this camera approaches 0 at an infinite object distance. When the object is closer, the negative distortion of the off-axis fields of view increases. The distortion ranges within ±2%.

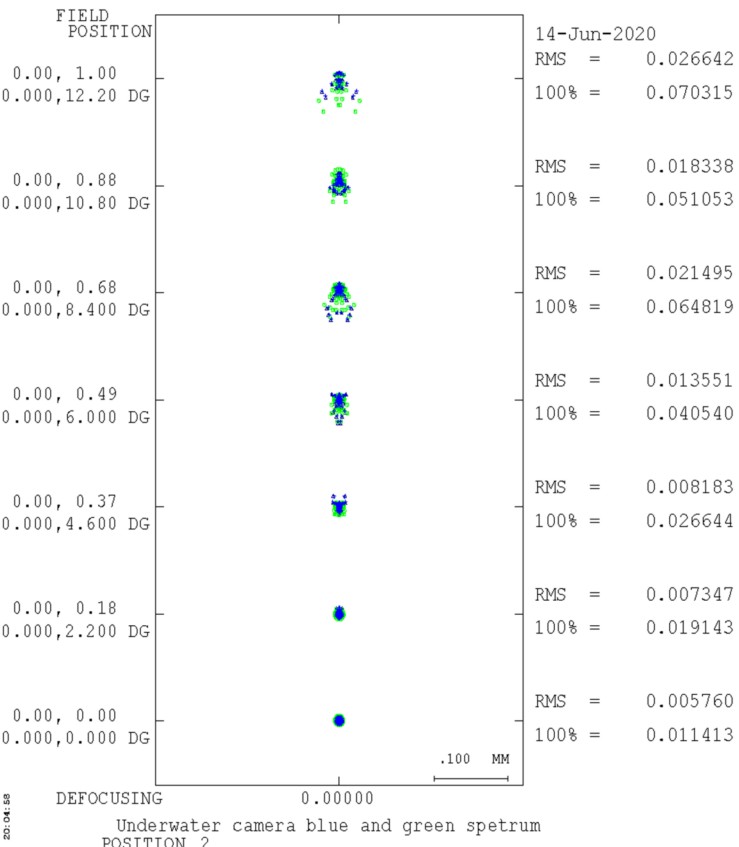

**Figure 12.** Spot size diagram with a 10 m object distance (generated by CODE V).

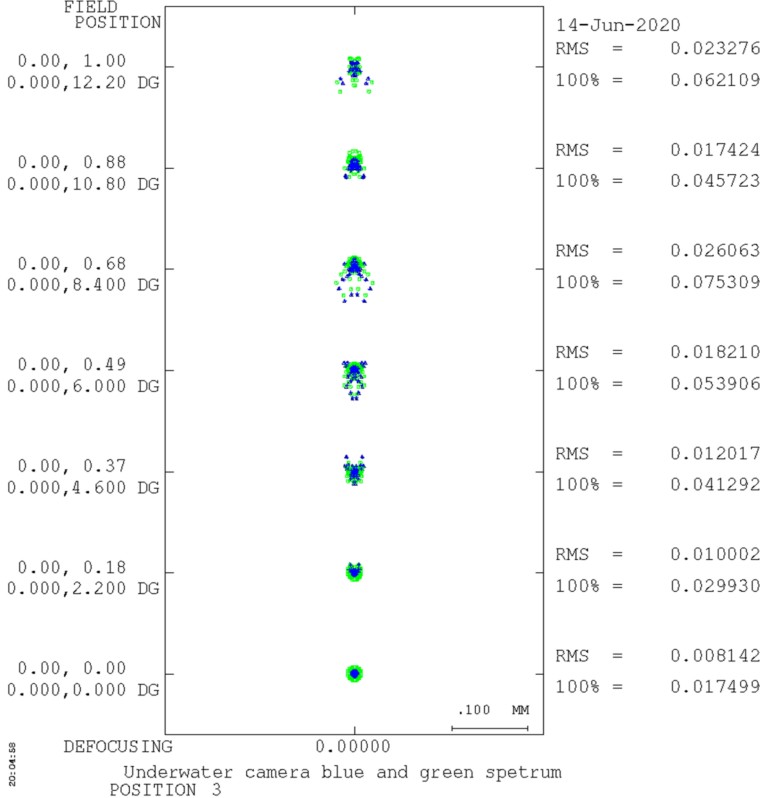

**Figure 13.** Spot size diagram with a 5 m object distance (generated by CODE V).

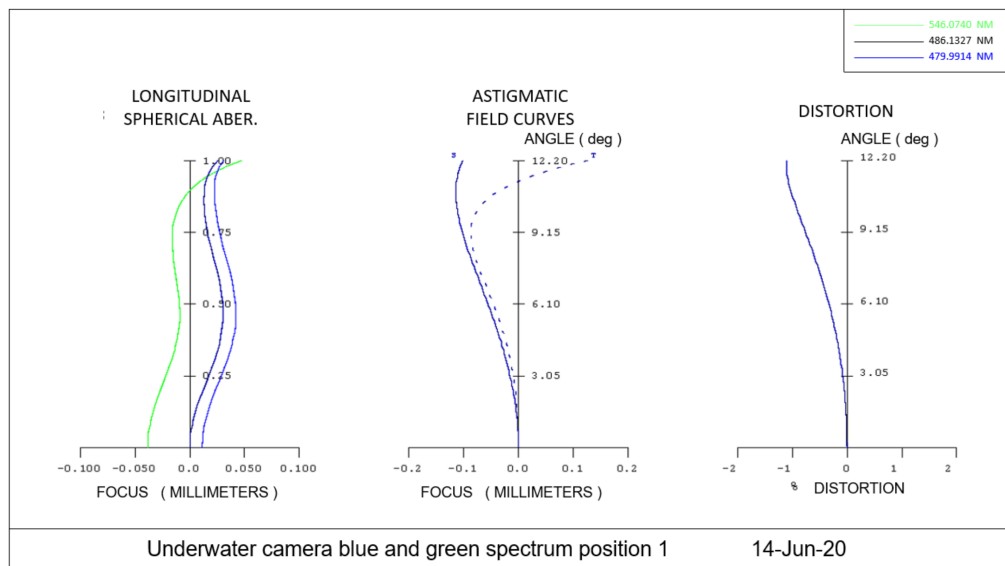

**Figure 14.** Optical aberration diagram with an infinite object distance ((**left**), axial chromatic aberration; (**middle**), astigmatic (blue line) field curvature (blue dot line); and (**right**), distortion).

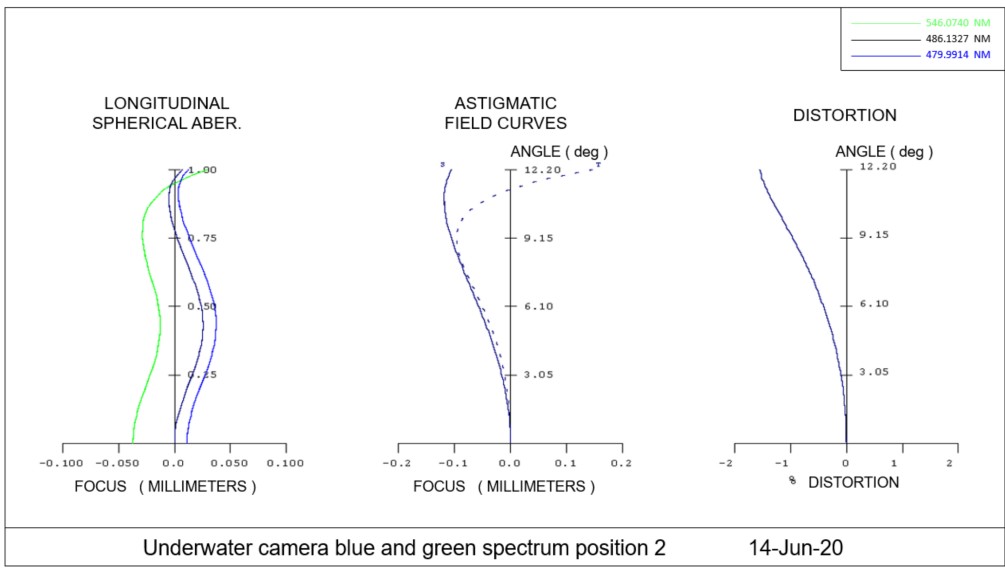

**Figure 15.** Optical aberration diagram with a 10 m object distance ((**left**), axial chromatic aberration; (**middle**), astigmatic (blue line) field curvature (blue dot line); and (**right**), distortion).

The modulation transfer function (MTF) is another method for inspecting the quality of an optical system. The horizontal axis is the spatial frequency, and the vertical axis is the modulation. The highest spatial frequency was defined as 70 lp/mm; the highest and lowest modulations were 1 and 0, respectively. Figures 17–19 are the MTF curves with an infinite, 10-m, and 5-m object distance. The outermost off-axis field of view was preferable with an infinite object distance and superior to the sixth field of view with an object distance of 5 m.

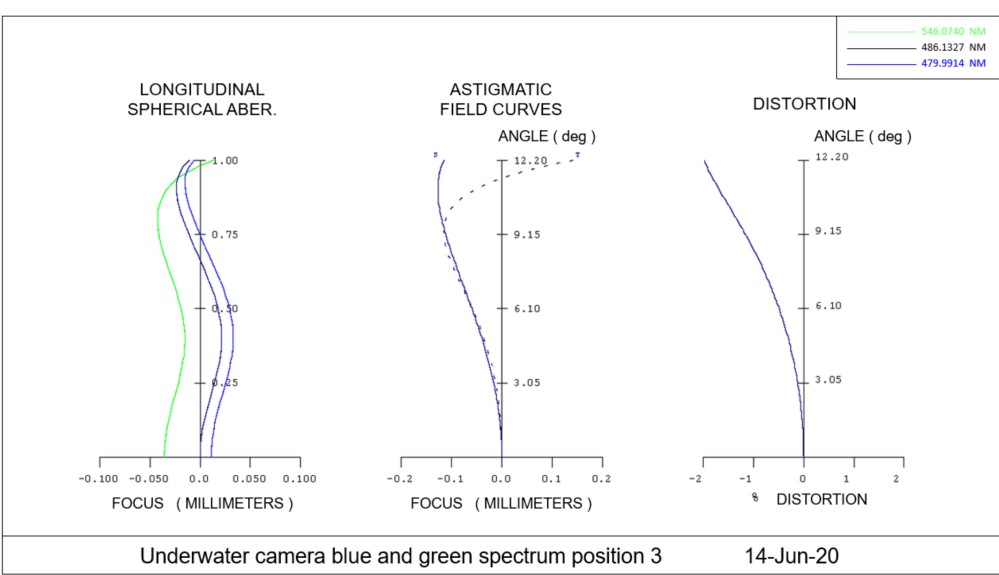

**Figure 16.** Optical aberration diagram with a 5 m object distance ((**left**), axial chromatic aberration; (**middle**), astigmatic field (blue line) curvature (blue dot line); and (**right**), distortion).

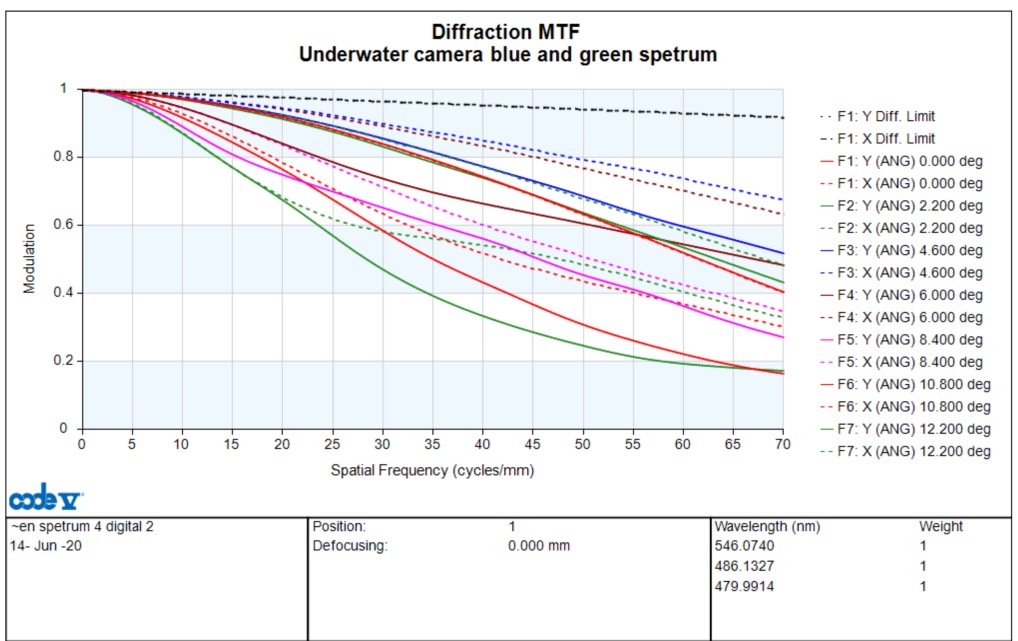

**Figure 17.** MTF curves with an infinite object distance.

Figures 20–22 were captured from two-dimensional simulations with object distances of infinity, 10 m, and 5 m. A picture we captured was used to simulate the results of the designed underwater lens by using CODE V. Because only blue light penetrates in the ocean, we added a filter that allowed only blue light to pass through to the first lens.

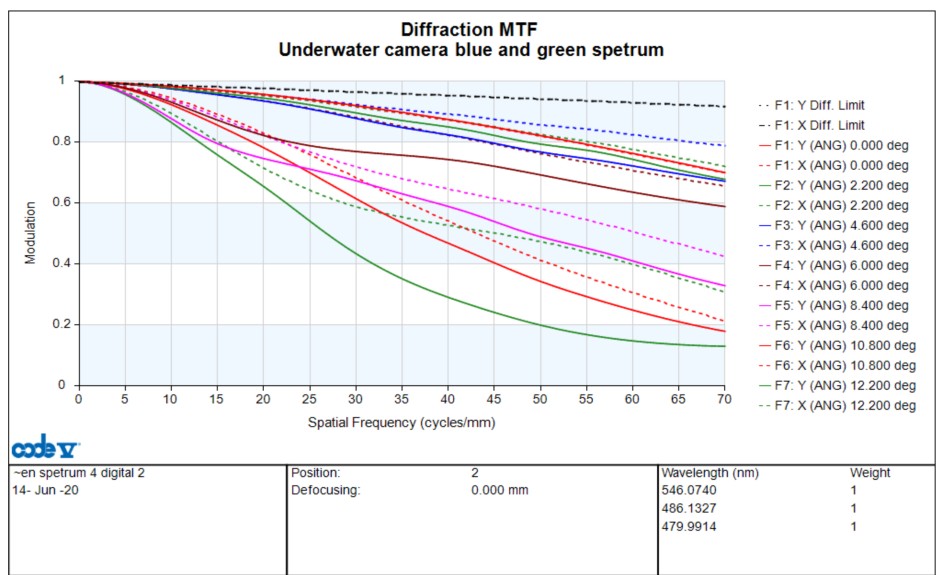

**Figure 18.** MTF curves with a 10 m object distance.

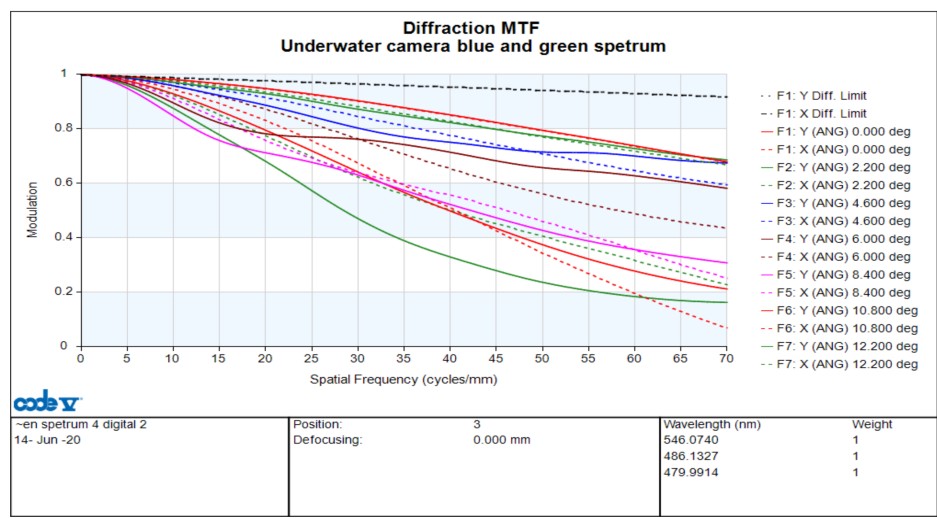

**Figure 19.** MTF curves with a 5 m object distance.

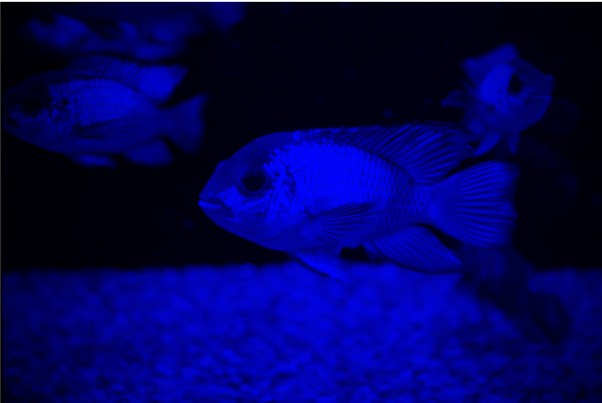

**Figure 20.** Two-dimensional simulation with an infinite object distance.

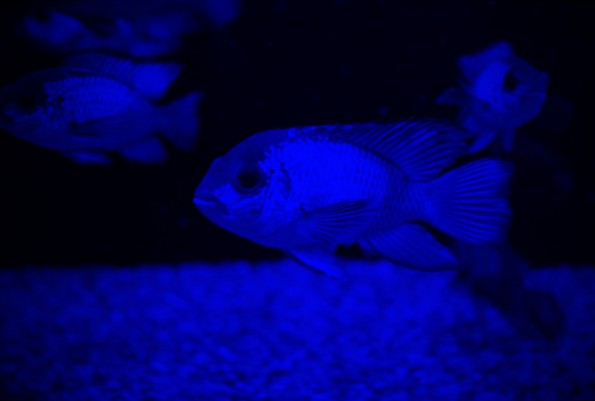

**Figure 21.** Two-dimensional simulation with a 10 m object distance.

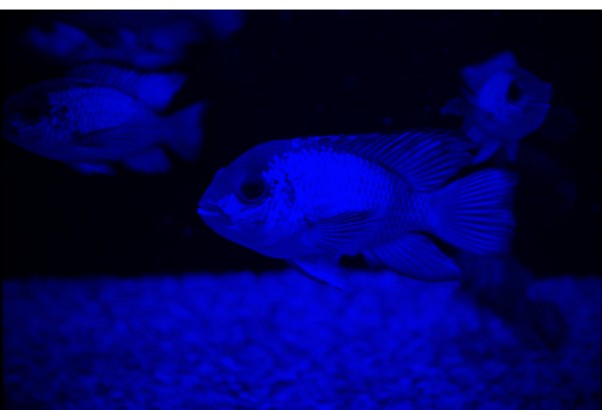

**Figure 22.** Two-dimensional simulation with a 5 m object distance.

## 5. Introduction to Genetic Algorithm and Its Extended Optimization Specific for Chromatic Aberration for Photogrammetry

Genetic algorithms [14,15] were used to define an objective function estimating the degree of compliance of different cameras with the specifications. In general, the defined objective function approaches an objective value of 0. When a camera obtains an objective value of 0, it complies with all requirements. The objective function of this study was defined as follows:

$$\text{obj\_Value (i)} = |\text{AXCL}| + |\text{LACL}| \tag{9}$$

where | | denotes an absolute value; and AXCL and LACL are the axial and lateral chromatic aberrations defined by Zemax, respectively.

Since the chromatic aberration is majorly related to the Abbe number of the material, the genes are to be the Abbe number of the lens elements for the GA optimization. There are 10 lens elements with glass material in the proposed lens so that the GA employs 10 genes for the chromatic aberration suppression. Subsequently, we randomly generated a population size (pop_size) with 10 genes (i.e., 10 Abbe numbers) to the designs of different lenses. The objective values of lens with different genes were obtained using the optical simulation software to evaluate the compliance of the lens with the specifications (that is, Equation (9)). The objective values were used to select crossovers. Many methods can be employed to select lenses for crossovers, among which the roulette wheel selection is the most common [16]. The proportion of the lenses on the wheel was allocated according to their fitness with the specifications. Lenses with superior fitness accounted for larger areas on the wheel; conversely, those with inferior fitness occupied smaller areas. However, the objective function we previously defined had an inverse relationship with the fitness value; that is, a smaller objective value was associated with a larger fitness value. Accordingly, we could not directly employ the objective value to select the lenses for crossover during

roulette wheel selection. Therefore, we defined the conversion relationship between the objective value and the fitness value as follows:

$$fit\_Value(i) = max\{obj\_Value(j) \mid 1 \leq j \leq pop\_size\} + min\{obj\_Value(j) \mid 1 \leq j \leq pop\_size\} - obj\_Value(i)$$
$$for\ i = 1, 2, \ldots, pop\_size \tag{10}$$

where the functions max{} and min{} are the maximum and minimum objective values in the pop_size lenses, respectively. In the simulation, we used probabilities to indicate the area of the wheel occupied by the lens sets according to their fitness value. Hence, the probability of the entire wheel is 1, and that of the lens sets could be determined using Equation (11).

$$p(i) = fit\_Value(i) / \sum_{n=1}^{pop\_size} fit\_Value(n)\ for\ i = 1, \ldots 2,\ ,\ pop\_size \tag{11}$$

The probability of the entire wheel is 1. We could obtain the area of each lens set on the wheel according to their fitness by using Equation (12).

$$q(i) = \sum_{n=1}^{i} p(n)\ \ for\ i = 1, 2, \ldots, pop\_size \tag{12}$$

and

$$q(0) = 0 \tag{13}$$

To randomly generate a number $\alpha$ between 0 and 1, if $q(i-1) < \alpha \leq q(i)$, we selected the genes of the $i$th lens for crossover. Moreover, the genes of two lens sets were required during a crossover; therefore, two selection mechanisms were required. Here, we let the genes of the two selected parent lens sets be $\mathbf{X} = (x_1, x_2, \ldots, x_{10})$ and $\mathbf{Y} = (y_1, y_2, \ldots, y_{10})$ and that of their children lens sets be $\mathbf{Z} = (z_1, z_2, \ldots, z_{10})$. The crossover procedure for generating the next generation was subsequently implemented after the genes of two parent lens sets were selected. Since both selected parents would propagate their features to the new genes, we further used random crossover to select the glass material as follows:

$$z_i = \beta x_i + (1 - \beta)y_i \quad for\ i = 1, 2, \ldots, 10 \tag{14}$$

where $\beta$ is a random number between 0 and 1, which indicated that the feature of new gene partially came from one of the parents and partially from the other. According to the preceding crossover calculation, we could obtain the genetics of the next-generation lenses. However, although living organisms adapt to their growth environment and pass on the better adapted genes to their next generation, their genes also mutate to make them more competitive in the environment. In terms of the application of the GA to the optimization of optical lenses, we defined a mutation rate $p_m$ between 0 and 1 and selected a random value between 0 and 1 for each gene. When $\gamma < p_m$, we implemented a mutation calculation for corresponding gene. Since the common Abbe number is from 25 to 80, the genetic changes of their subsequent generation are expressed as follows:

$$z_i = random\ number\ from\ 25\ to\ 80 \tag{15}$$

In the preceding selection, crossover, and mutation were the basic calculations of the genetic algorithm. Figure 23 presents the implementation procedure of the GA in this study. This GA is based on pop_size 200, crossover rate 0.8, and mutation rate 0.3 for running 200 generations.

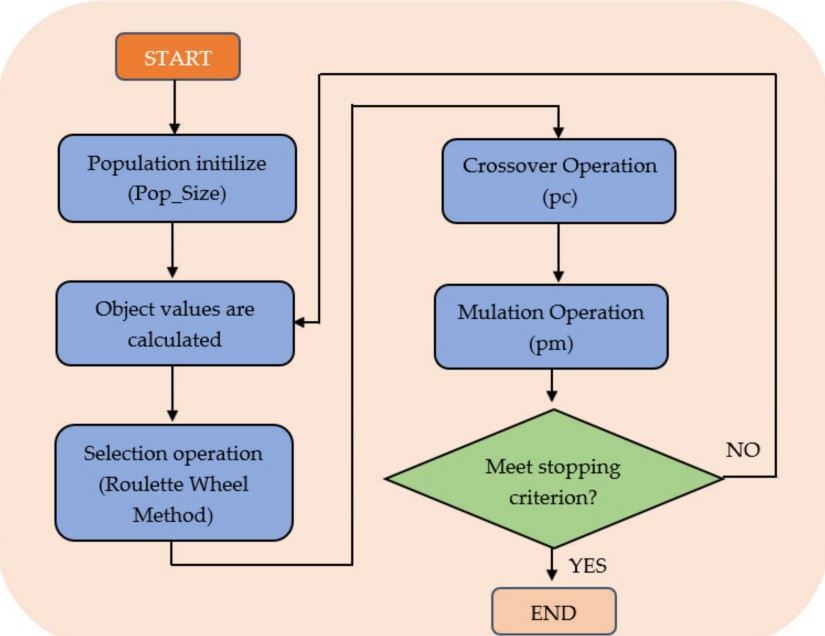

**Figure 23.** Flowchart of the genetic algorithm calculation.

## 6. Methodology for Extended Optimization of Zoom Optics with Genetic Algorithm Specific for Chromatic Aberration

Optimization was executed by CODE V basically for most aberrations in this system so that it played the role at modulation transfer function (MTF). Genetic algorithm in this research was close to so-called "extended optimization" specific for chromatic aberrations only and nothing to do with other aberrations in this system. Therefore, there was no obvious contribution to MTF (2%) of the whole system via genetic algorithm extended aberrations.

Why is chromatic aberration so sensitive to lens for under water purposes although it might not play the role at system MTF? Chromatic aberration might cause focus shift due to various wavelengths. From 1 m to 100 m for under water applications, there is plenty of variety regarding wavelengths so that is a better choice to minimize chromatic aberration first in order to guarantee the best performance from 1 m to 100 mm for under-water image application.

From the simulation of CODEV and Zemax, there might be only 2% improvement of simulation MTF. Why is chromatic aberration so critical in practical systems? The key point is that chromatic aberration might be strongly-emphasized with so called integral aberrations effects due to ocean currents under water, which can be seen as a kind of time-domain variance of the glass index, which seriously degrades system performance although CODE V or Zemax is not able to simulate that cases. So far, there is no similar software or digital signal processing to overcome this kind of "ocean aberration" but minimization of chromatic aberration might be successful in minimizing such types of time-domain aberrations; similar optical software is used in the air in order to overcome air turbulence.

In this section, we propose an extended optimization method via genetic algorithm specific for chromatic aberration applied to photogrammetry. This extended optimization method will efficiently eliminate the chromatic aberration although it might be not of help in improving MTF. The optical design of zoom lenses for underwater is different from that of prime lenses. First, changes in light spectra increase the chromatic aberration and focus-point displacement for zoom lenses compared with prime lenses. The various optical aberrations generated by zoom magnification are difficult to control. Second, zoom lenses generally have lower resolving power than prime lenses and produce more complex optical aberrations with changes in magnification [17]. Therefore, designing a zoom lens with a

high resolving power for underwater photography and the ability to adjust with changes in water depth is difficult.

In accordance with Polaris specifications, we developed a zoom lens with three times zoom ratio. The specifications are listed in Table 6. The half-angles of view and their weight ratios are listed in Table 7. All optical layouts are demonstrated on Figure 24 optimized by Zemax with genetic algorism. Its MTF performance is illuminated by Figures 25–27 according to difference zoom ratio 5 mm, 10 mm, and 15 mm.

**Table 6.** Specifications of the novel underwater zoom lens.

| | |
|---|---|
| Resolving power | 2100 K pixels |
| Sensor size | 12.8 mm × 9.6 mm (CMOS) |
| Diagonal length | 14.6 mm |
| Actual image height | 7.3 mm |
| Object distance | Infinity |
| Spatial cutoff frequency | 70 lp/mm |
| Angle of view | 72.20° |
| Focal length | 5 mm to 15 mm 3x |
| Aperture | f/2.8 to f4.5 |

**Table 7.** Field of view and weights.

| Angle of View | Weight |
|---|---|
| 0° | 3 |
| 4.6° | 3 |
| 9.1° | 2 |
| 18.2° | 2 |
| 27.3° | 2 |
| 36.7° | 1 |

Zoom optics has its inherent difficulties in optical performance compared to prime lenses especially in astigmatism, distortion, and MTF performance. In this case, fixed overall length complicates optical design and optimization so that its performance is not as good as fixed focal optics. There are some alternatives to improve the performance of zoom optics although their performance is not equivalent to optics with fixed focal length. Generally speaking, one is the employment of very-high-index optical glass and another is employment of more optical elements. Unfortunately, both methods are not feasible in this specification from Polaris as optical glass with high index might have poor physical and chemical characteristics so that it is not a good choice for optics under water and employment of more optical glass will make optics longer than expected, which is also not good for optics under water.

Despite the two concerns of optical aberrations and resolving power, zoom lenses have more degrees of freedom than prime lenses. Therefore, we used the genetic algorithm for optimization and conducted optical design in Zemax as follows: the optimal glass lens sets were initially selected in Zemax based on the genetic algorithm to significantly reduce chromatic aberration, particularly axial chromatic aberrations [18,19]. Various approaches can be used to reduce axial chromatic aberration; however, the most direct and efficient method is to select an appropriate optical glass. Neither the least damping square in Zemax nor CODEV could reduce the chromatic aberration of the zoom lens to a low value [17,20]. However, we incorporated the high-level language of the genetic algorithm into Zemax for optimization and obtained a set of lenses that produced relatively low axial chromatic aberration. We made minute adjustments to the zoom lenses and further identified the

displacement of the highest resolving power lenses separately at different depths and under different spectra.

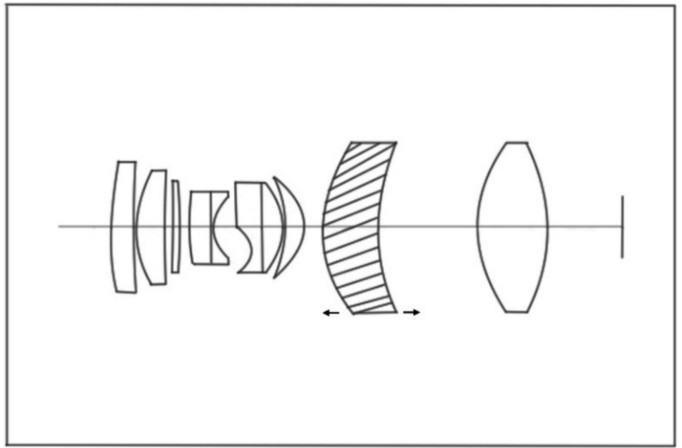

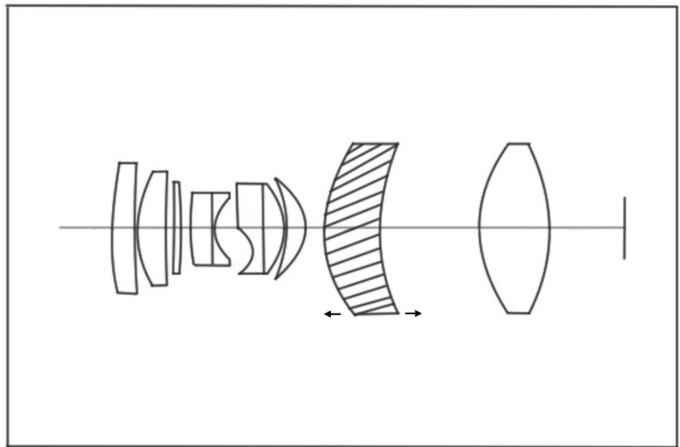

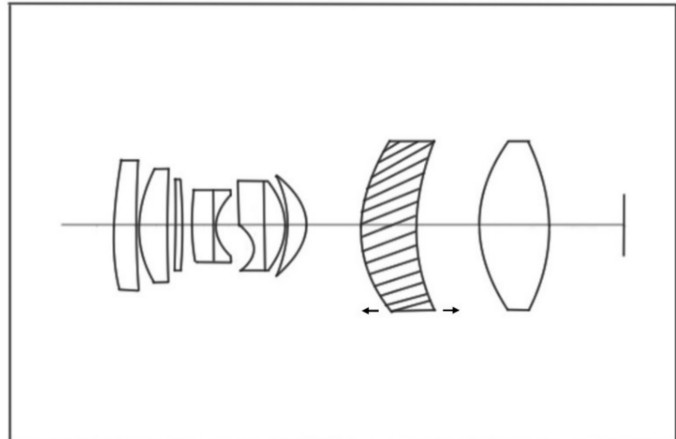

**Figure 24.** Optical layout of underwater optical zoom optics. From top to bottom, the focal length is 5 mm, 10 mm, and 15 mm. This zoom design is characterized with only one element for zoom function, which significantly simplifies opto-mechanical system design for underwater image systems.

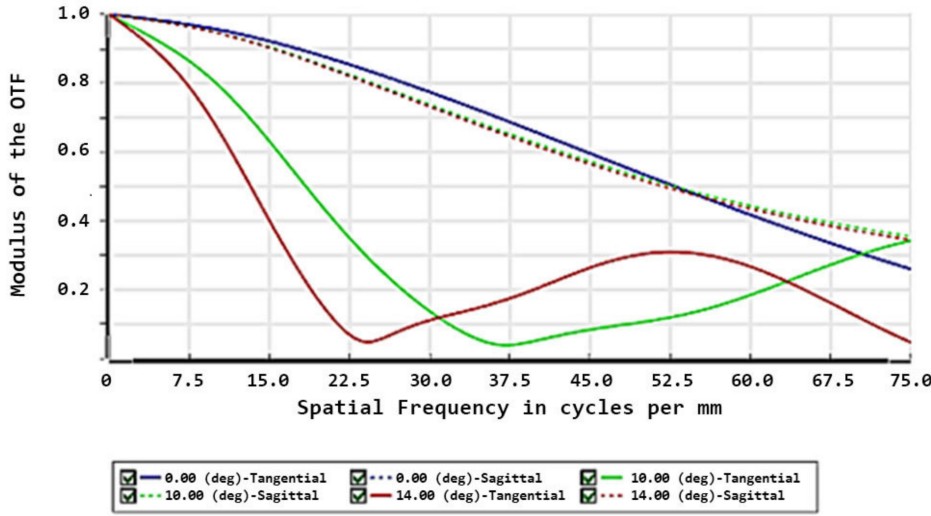

**Figure 25.** MTF chart for zoom 1 (focal length = 5 mm).

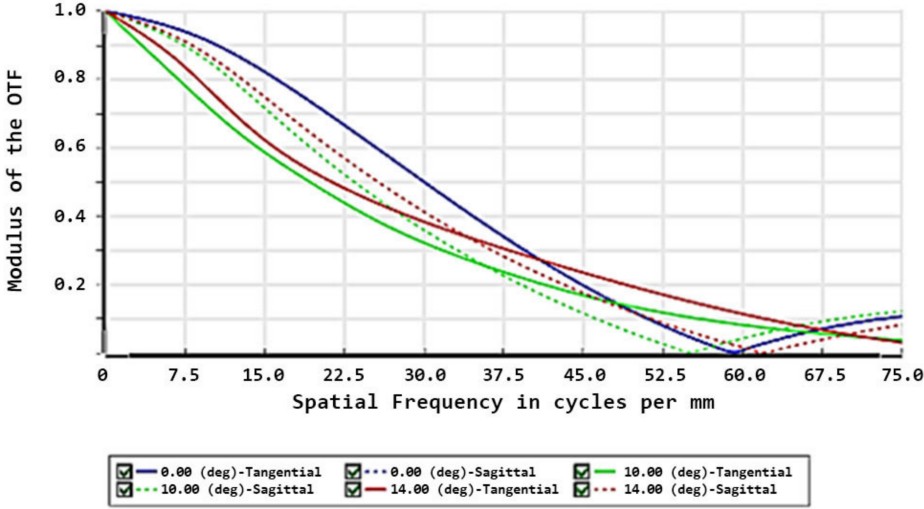

**Figure 26.** MTF chart for zoom 2 (focal length = 10 mm).

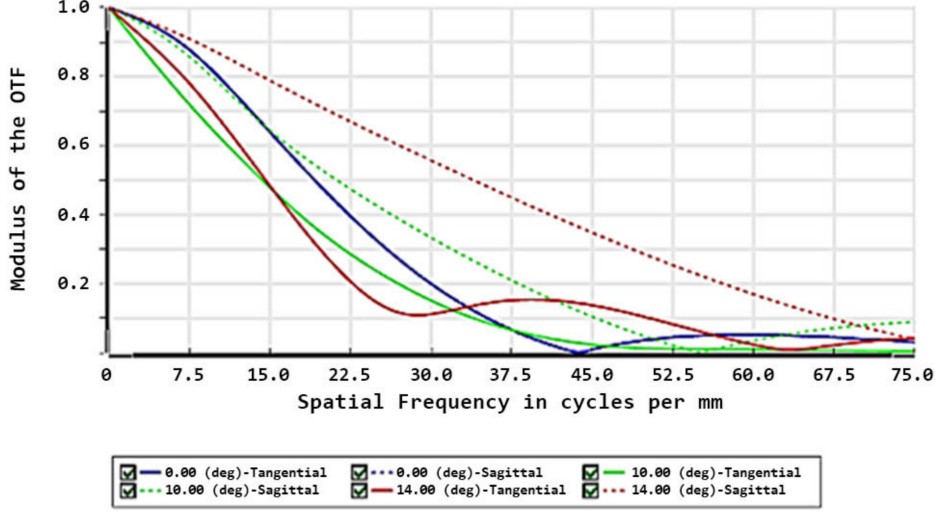

**Figure 27.** MTF chart for zoom 1 (focal length = 15 mm).

The reason why it is very difficult to compare volumetrically this optical design with conventional optical design is that the lens design in this paper is without a dome. Compared with traditional optical components, the proposed zoom exhibits higher MTF and provides a more satisfactory image quality. Because of the advantages of the proposed MTF zoom, which are difficult to achieve for a traditional single lens, it has the potential to be applied to in-camera systems and virtual and augmented reality. For example, the medium index outside the lens given in this paper is different from that of general optics with a dome. Their boundary conditions after optimization are also completely different. From the point view of optical measurement, there will be a flat glass in the front of conventional optics with dome, which might degrade MTF up to 20% or more. Both are, however, made in different worlds so that it is very difficult to find a basic standard with which to compare them.

## 7. Discussion and Conclusions

In accordance with the requirements of the survey vessel Polaris, we proposed a novel optical design with prime and zoom lenses and a high resolving power in the full light spectrum in shallow water and the blue–green light spectrum in deep water. We specifically applied a genetic algorithm and Zemax software to optimize a 3x zoom lens for use in deep water with a minimized chromatic aberration value. This camera can prevent sensitive chromatic aberration and maintain its imaging quality during underwater surveying when spectra change.

Both the prime and zoom lenses achieved the following experimental results:

First, the developed lenses enabled independent operation without an airtight box. In the past, cameras were placed in waterproof airtight boxes for underwater photography, which caused problems such as chromatic aberration and focus-point displacement and made deep-water photography difficult. Therefore, we proposed an optical design for both prime and zoom lenses. The prime lens had a focal length of 75 mm and an aperture of f/2, and the zoom lens had an effective focal length of 5–15 mm and an aperture of f/2.8–f/4.5. Conventional waterproof boxes were not used this optical system. We adopted a waterproof dome window on the outer surface as the basic structure and optimized it in accordance with different water depths and light spectra to obtain a distortion within ±2% and high-resolution underwater imaging quality.

Second, both the prime and zoom lenses could reach at least a 20% increase in resolving power at a cutoff frequency of 70 lp/mm compared with conventional cameras with waterproof doom.

The zoom lens was optimized using the genetic algorithm and designed to be applicable at all water depths. Compared with conventional underwater cameras, its volume and weight were reduced by more than 50% and 60%, respectively, and its resolving power was increased by 30–40%.

**Author Contributions:** C.-F.C., C.-M.T., C.-H.C., Y.-H.W. and Y.-C.F., carried out the experiment and wrote the manuscript with support from H.-Y.L. (Hsiao-Yi Lee) and H.-T.L. S.-H.C. fabricated the sample H.-Y.L. (Hsing-Yuan Liao) helped supervise the project. Y.-H.W. and Y.-C.F. conceived the original idea, supervised the project. C.-C.W. is the head of Polaris vessel in this research. His contribution is related to funds and supervision. All authors have read and agreed to the published version of the manuscript.

**Funding:** This study was part of the Ministry of Science and Technology project number MOST108-2221-E-992-088, MOST 109-2221-E-005-074, MOST 109-2218-E-005-012, MOST 110-2221-E-005-054-MY2, MOST 109-2221-E-005-074 and the 2019 Ocean Characteristic Cross-Campus Research Project (No. 108D12) and 109E9010P02. We thank the Ministry of Science and Technology and the Office of Marine Science and Technology, National Kaohsiung University of Science and Technology, for their support of this study.

**Institutional Review Board Statement:** Not applicable.

**Informed Consent Statement:** Not applicable.

**Data Availability Statement:** Due to the nature of this research, participants of this study did not agree for their data to be shared publicly, so supporting data is not available.

**Conflicts of Interest:** The authors declare that they have no conflict of interest.

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
