# Peer review of "Optical Design and Optimization with Genetic Algorithm for High-Resolution Optics Applied to Underwater Remote-Sensing"

_applsci, doi:10.3390/app112110200_

Round 1

Reviewer 1 Report

This paper is suitable for publication in the Applied Science Journal.

However, the author has to improve table 1, correct some typos error such as in lines 127, 289, etc.

Author Response

Please find the file attached. Many thanks for your kind instruction.

Reviewer 2 Report

In manuscript the underwater camera with prime and zoom lenses and a high resolving power was developed. The camera can be used in shallow water and in deep water.

After review of manuscript I have following questions.

  1. What explains the need to use Zemax and Code V CAD systems of the optical systems to synthesize a camera? Why isn't chromatic aberration correction done in Code V?
  2. It is not clear which components are used to zooming. It is necessary to substantiate the zoom structure, in which to show the groups of movable and fixed elements of the optical system. It would be nice to show them in Figure 10 or a separate figure.
  3. The optical systems in Figure 10 and Figure 24 are different. However, the analysis of the test image quality at different depths is carried out for the optical system in Figure 10. This needs to be justified. it is desirable to write the design parameters of the camera.
  4. It is not clear how it is determined that the volume and weight of the developed chamber are reduced by more than 50% and 60%, respectively, compared to other cameras.

There are the following remarks on the formatting.

  1. In the heading of the columns of Table 1, words should be formatted without a line break.
  2. Formulas 2-8 are given, but it is not entirely clear how they are used in the calculations.

Author Response

Please find the file attached. Your kind assistance are much appreciated.

Reviewer 3 Report

The submission describes the design and optimization of the optics of an underwater camera. The idea to explicitely design the optics for underwater, especially by taking the shifts in the spectrum from shallow to deep sea into account, is very nice and worth publishing. The approach is very thorough and leads to convicable results.

The main drawback is the presentation. There are many language glitches and bugs. I am not a native speaker myself and I refrain from listing them all. Also, many figures are small and require that the readers zooms into the PDF. The organization of the article is OK. The title should be revised: (a) there is no mentioning that this is about an underwater application in the title; (b) mentioning photogrammetry there is not ideal as that is not the main focus of the article; (c) even there as language bugs ("algorism").

Concluding, it is very nice work but the language and presentation must be polished.

Author Response

Please find the file attached. Your kind instructions are much appreciated. 

Reviewer 4 Report

1) First of all, This paper did not provide any evidence about your scientific accomplishment.  Just Other people's comments. In the offshore market, there are so many underwater zoom cameras like Simrad. The introduction phase must describe previous work or related products and the optimization methods.

2) You must explain the cause that you choose the genetic algorithm.

3) All figure's resolution is not fine. I can't understand each figure's contents. Especially, Figures 20,21,22, I guess these photos are important in your method, But I can not distinguish each figure's difference.  

Author Response

Please find the file attached. Many thanks for your kind assistance and instruction. 

Round 2

Reviewer 2 Report

After review of manuscript-version 2 I have following questions.
There are no references on the zoom calculation in the introduction and in the text.
it would be nice to make a reference to the literature to formulas (2) - (8).
In fig. 24 depicts a component to the left of a movable component with an asymmetrical first surface. Is it really so? In Figure 10, this component has a symmetrical surface.
In my opinion, in "Discussion and Conclusion" the conclusion about the smaller mass and volume of the developed optics in comparison with other cameras is not sufficiently substantiated. It is not clear which cameras are being compared with.

Author Response

Please find the file attached.
